# Cyclic Shear Behavior of Frozen Cement-Treated Sand–Concrete Interface

**DOI:** 10.3390/ma15248756

**Published:** 2022-12-08

**Authors:** Rongkai Pan, Zhaohui (Joey) Yang, Ping Yang, Xin Shi

**Affiliations:** 1School of Civil Engineering, Nanjing Forestry University, Nanjing 210037, China; 2Department of Civil Engineering, University of Alaska, Anchorage, AK 99508, USA

**Keywords:** cyclic shear behavior, frozen cement-treated sand, shear stress, normal displacement, shear stiffness

## Abstract

The cyclic shear behavior of frozen cement-treated soil–concrete interfaces is critical for analyzing soil–structure interfaces and foundation design in cold regions and artificially frozen ground. The cyclic shear behavior of the interface between frozen cement-treated sand and structure is investigated in this paper at various normal stresses and temperatures. Experimental results include the variation of the peak shear stress, peak normal displacement, shear stiffness with the number of cycles, and the relationship between peak shear stress and smoothness under certain conditions. Peak shear stresses of warm frozen cement-treated sand and cold frozen cement-treated sand varied with cycle number. Additionally, the former is significantly larger than the latter in the stable phase. The peak normal displacement showed the same results, indicating that the ice crystals formed on the surface and the strength of the frozen cement-treated sand have significant differences at various temperatures. The study’s findings aid in understanding the complexities of the cyclic shear behavior of frozen cement-treated sand and structure interfaces and provide references on frozen cement-treated sand zones in practical engineering.

## 1. Introduction

The interface between frozen soil and structure is critical for the safety of underground space construction in artificially frozen ground and cold-region infrastructures such as subway tunnels [1], building and bridge foundations, freeways, embankments, and oil pipelines [2,3,4,5]. However, due to the significant difference in mechanical properties between frozen soil and structure, the interfaces can become fragile during earthquakes or wind [6,7,8], and engineering accidents are likely to occur. Therefore, engineering studies of the mechanical and deformation properties of frozen cement-treated soil and structure interfaces are critical.

Researchers have studied the mechanical properties of unfrozen soil and structure interfaces in various ways, including laboratory experiments [9,10], finite element analysis [11,12], and theoretical analysis [13,14]. The cyclic direct shear behavior of the soil and structure interface has also been studied. Chenari et al. [15] studied the properties of the sand and expanded polystyrene (EPS) mixture interface and geogrid reinforcement under cyclic loading. They performed a series of cyclic tests and investigated the effects of normal stresses, cyclic shear amplitudes, and cycle numbers. Zhang et al. [16] developed a modified elastoplasticity damage model to account for the monotonic and cyclic behaviors of the geotextile-gravelly soil interface. They developed new damage variables and a shear strength criterion based on the test results. Vieira et al. [17] studied the behavior of the silica sand and high-strength geotextile interface under cyclic loading conditions. They reported that the interface stiffness tends to increase during the first loading cycle; however, it exhibits slight variation after ten cycles.

Therefore, when cyclic loading is applied, the characteristics of an interface are prone to change, and stress degradation may occur under certain conditions. Liu et al. [18] investigated the cyclic attenuation of shear/normal stress amplitude, strength reduction, and particle crushing in depth in the associated pile and sand interface cyclic weakening mechanism. Ghionna et al. [19] investigated the behavior of sand and structure interfaces under cyclic loading. The results showed that cyclic loading, which increases the contractivity of the interface, causes the sand-plate frictional resistance to degrade. Mortara et al. [20] investigated stress degradation at sand and structure interfaces using a constant normal stiffness direct shear apparatus.

The shear behavior of the frozen soil–structure interface is significantly different from that of the unfrozen soil and structure interface. The frozen soil and structure interface has been studied via field and indoor tests [21,22,23]. Biggar et al. [24,25] also provided the design information for the Short-Range Radar sites in Canada via a pile load test program. Furthermore, the effects of temperature and salt content on the adfreeze strength were examined by in-situ model tests [26]. Since the interface may be of made different materials, various materials for the interface have also been investigated [27,28,29].

Furthermore, test results are greatly affected by the size of the specimen. He et al. [30] investigated the shear behavior of frozen clay and concrete interfaces using a large-scale shear device. They discovered that increasing the initial water content and decreasing the temperature or ice content increases the growth rate of vertical displacement, which is due to the ice film’s effect on particle size. Shi et al. [31] investigated the sub-peak adfreezing strength using a large-scale frozen soil and structure interface shearing instrument, and a three-element sub-peak adfreezing strength prediction model coupled with interface temperature, normal stress, and roughness was built using multivariate nonlinear regression. Liu et al. [32] created a large-scale direct shear test system to investigate the shear behavior of the frozen coarse-grained soil and concrete interface. They discovered that peak shear strength had linear relationships with normal pressure and temperature, but a nonlinear relationship with water content had a much larger effect on peak shear strength.

In the study of the cyclic shear behavior of frozen soil–structure interface, Zhao et al. [33,34,35] used a large-scale direct shear apparatus to study the cyclic shear behavior of frozen soil and structure interface under various conditions, such as constant normal stresses, sub-zero temperatures, and constant normal stiffness. They analyzed shear stress, normal displacement, friction angle, cohesion, and dilation and proposed a simple damage model to describe the behavior of the frozen soil–structure interface. Wotherspoon et al. [36,37] tested identical full-scale column-foundation systems during summer and winter to demonstrate that freezing temperatures and a thin layer of frozen soil can significantly influence the soil-foundation-structure interaction (SFSI) and lateral load response of bridge columns supported by deep foundations.

However, in practice, the soil is commonly reinforced with cement to ensure the safety of construction, and then the cemented soil freezes due to artificial or natural factors. For example, deep soil mixing is often used to strengthen the origination and reception areas of the shield, and then the ground freezing method is applied [38,39]. Meanwhile, the surrounding soil of many buildings, bridges, and tunnels in cold regions requiring cement improvements [40,41] also forms the frozen cemented soil after the temperature drops to sub-zero temperatures.

In summary, the shear behavior of the frozen soil and structure interface has been extensively studied. However, the cyclic shear behavior of the frozen cement-treated soil and structure interface, particularly the behavior of warm frozen cement-treated soil and cold frozen cement-treated soil and concrete interface, was ignored. Warm frozen soil is susceptible to temperature changes, so a significant reduction in bearing capacity can easily lead to risks in related projects. Therefore, attention must be paid to deformation and damage. However, it is important to note that the temperature range of warm frozen soil is not clearly defined. In this study, warm frozen cement-treated sand and cold frozen cement-treated sand are defined as cement-treated sands with temperatures greater than and less than −2 °C, respectively, based on the characteristics of the results.

The cyclic shear behavior of the warm frozen cement-treated sand- and cold frozen cement-treated sand–concrete interface at various normal stresses and temperatures is investigated in this study. Experimental results, including peak shear stress, peak normal displacement, shear stiffness variation with the number of cycles, the relationship between peak shear stress and smoothness, and the different interface shear behaviors of frozen sand and frozen cement-treated sand, are analyzed.

## 2. Materials and Methods

### 2.1. Test Soil, Cement, and Specimen Preparation

The typical Nanjing sand was used to prepare the specimens. Figure 1 depicts the grain-size distribution, and Table 1 lists the test soil parameters. The Unified Soil Classification System (SP) has classified it as poorly graded sand. According to engineering standards, the cement slurry was created by thoroughly mixing water and Portland slag cement (Grade 32.5) in a 1:1.5 ratio. The cement content is 15%, and the weight ratio of dry cement and reformed sand. The water content is 27% according to natural sand, and the sand was prepared by thoroughly mixing dry soil and water. The sand and the cement slurry were thoroughly mixed and then filled into the molds to prepare cement-treated sand specimens. Subsequently, these molds were placed in a curing room, with a temperature and relative humidity of 25 °C and 97%, respectively. The mold was removed after three days, and the surface of the specimen was smoothened. Finally, the specimens were placed in the curing room for an additional 11 days to meet the 14-day curing requirement [JGJ/T-233, 2011].

### 2.2. Large-Scale Direct Shear Apparatus

Figure 2 depicts the use of a large-scale direct shear device, the DDJ-1G, to investigate the cyclic shear behavior of the frozen soil and concrete interface. The loading system provides horizontal and vertical loads and can simulate constant stress tests. It can perform cyclic and monotonic shearing and imitate interface roughness of different types by changing the shear plates. The inner box, which is 200 mm long × 100 mm wide × 55 mm tall, is used as the specimen container and is placed inside the shear box. The device can test specimens at precisely controlled temperatures of −20 °C to +30 °C through refrigeration and a temperature control system. After installing the displacement and temperature sensors, circulation channels, and environmental chamber, the cooling system was activated. The cooling capacity was transferred to the specimen via the shear box and shear plate. The temperature sensor is inserted into the specimen through three holes on the top of the shear box, as shown in Figure 2b (10). When the specimen reached the specified temperature (in about three hours, according to the refrigeration effect verification before the test) and was allowed to achieve thermal equilibrium for one hour to ensure the maximum temperature difference within the specimen was less than 0.2 °C, normal stress was applied and allowed to stabilize for another two hours before beginning the direct shear test; the total freezing duration was six hours. After the temperature and displacement are stabilized, the horizontal load is applied. The shear plates are made of steel. Previous studies have reported that the shear behavior of the interface is significantly affected by roughness [1,33,42]. The surface of the shear plate consists of a series of grooves and flat intervals between them, as shown in Figure 2b. The depth of the grooves is defined as roughness R. Since roughness of 0.8 mm well simulates the surface of concrete, it was adopted in this study. Following the tests, the inner shear box was removed and examined under the microscope. Photographs of the specimen’s surface were taken at various magnifications to study the surface variation.

### 2.3. Experimental Schedule and Procedure

As shown in Table 2, the testing parameters were eight temperatures and four normal stresses. The cyclic shear amplitude and shear rate were set at 5.5 mm and 5 mm/min, respectively. The number of cycles was 30, and the tests ended as the target number of cycles N was reached.

Figure 3b depicts the temperature oscillations for testing temperatures of −1 °C and −14 °C. The amplitudes of the temperature oscillations are seldom greater than ±0.1 °C, although there are some fluctuations. As shown in Figure 3b, it can be observed from the shear displacement control during testing that the system works well.

## 3. Results

### 3.1. General Observations

The test observed two types of variation in shear stress and normal displacement with the number of cycles at various interface temperatures. The 25 °C, −1 °C, −1.5 °C, and −2 °C show similar types of variation, while −4 °C, −6 °C, −10 °C, and −14 °C show significantly different kinds of variation. Chai et al. [43] found that when the cement content was 15%, the unfrozen water content of cement-treated soil significantly decreased with decreasing temperature. Therefore, the ice crystals increased with decreasing temperature, resulting in a significant change in the interface, thus affecting the shear behavior. The variation curves of shear stress and normal displacement during testing at −1 and −14 °C were chosen for display. Figure 4 depicts the typical interface shear stress and normal displacement with the number of cycles at −1 °C and −14 °C under a normal stress of 700 kPa. During testing, the peak shear stress of each cycle shows no discernible variation at −1 °C. The normal displacement increases significantly with the increase in the number of cycles. However, at −14 °C, the maximum shear stress appears in the initial stage of the first cycle, then it decreases sharply and fluctuates little in the subsequent cycles, which is defined as the stable phase in the test. The normal displacement shows a significant increase in the first few cycles but slowly increases in the subsequent cycles, forming a stable phase. Peak shear stress is a critical factor in shear behavior. As shown in Figure 4, a curve is drawn at the point of the peak shear stress in each cycle to facilitate the comparison of the shear behavior under different temperatures and normal stresses. The forward peak shear stress and the absolute value of the reverse peak shear stress are represented by the black and red colors, respectively. When the horizontal loading device applies a force to the right, the shear stress is forward, and the shear stress is reversed when the force is applied to the left. Similarly, a curve is drawn with the peak normal displacement of each cycle to analyze the variation of the normal displacement.

### 3.2. Variation of the Peak Shear Stress with the Number of Cycles at Various Conditions

The relationships between the peak shear stress and the number of cycles for the frozen cement-treated sand–concrete interface in various conditions are shown in Figure 5. The shear stress obtains two different peak values within one cycle, indicating that the shear stress of the interface is anisotropic. Furthermore, it is due to the shear orientation effect, which is the macroscopic manifestation and response of the cycle for cement-treated sand due to shearing near the shear plate.

The variation of peak shear stress of warm frozen cement-treated sand under different normal stresses is significantly different, while the forward and reverse peak shear stresses under the same normal stress are similar. The forward and reverse peak shear stresses of cold frozen cement-treated sand differ initially but then change similarly. Furthermore, the peak shear stresses will increase or decrease in the first few cycles at various normal stresses, showing a trend of differentiation in the stable phase. Warm frozen cement-treated sand in the stable phase has close peak shear stresses to cold frozen cement-treated sand, and the former is significantly larger than the latter. Generally, the rate at which the shear stress reaches the stable phase increases as normal stress increases, indicating that friction is vital. In the stable phase, the peak shear stress is primarily due to the friction between the ice crystals at the structural panel and cement-treated sand interface since increased normal stress damages the specimen surface in each cycle and decreases friction gradually, resulting in no apparent change in the peak shear stress after the maximum damage is reached.

At −2 °C, there are a few ice crystals on the surface of the specimen, the ice bondage formed in the interface is small, and the strength of the specimen is less than that of the cold specimen. Therefore, the peak shear stress has three variations at different normal stresses. Because the ice bondage is greater than the friction at 100 kPa, the shear stress has a prominent peak in the first cycle, then drops sharply and gradually stabilizes. At 300 kPa, the peak shear stress first falls, then rises and gradually stabilizes, which can be attributed to the change of the frozen cement-treated sand at the interface during cyclic shear. The frozen cement-treated sand on the specimen’s surface is gradually compacted and smoothed during the first few cycles, reducing sliding friction. In the subsequent cycles, it is continuously broken due to the normal stress, which increases the sliding friction gradually. At 500 kPa and 700 kPa, the peak shear stress is minimal for the first cycle, and then it gradually rises and reaches the stable phase. Since the frozen cement-treated sand is broken due to the excessive normal stress, friction gradually increases.

It is to be noted that the maximum shear stress of cold frozen cement-treated sand appears in the first cycle and decreases sharply after that. The shear stress eventually reaches a stable phase except for −4 °C and 700 kPa, where the peak shear stress gradually increases after reaching the minimum. Therefore, −4 °C is considered the critical point of the cyclic shear behavior of the frozen cement-treated sand interface since the normal stress does not exceed 700 kPa.

The difference between the peak shear stress in the forward and reverse directions under typical conditions can describe the anisotropy extent of peak shear stress. The variation of the anisotropy extent with normal stresses and temperatures at N = 1 is shown in Figure 6. With increasing normal stress and decreasing temperature, the anisotropy extent disperses. Furthermore, for small normal stress, the frictional restraint effect of the shear plate on the particles is small, and the shear orientation effect is weak, resulting in a low dispersion of the anisotropy extent of the peak shear stress. However, for increased normal stress, the restraint effect of the shear plate is strengthened, and the shear orientation effect of the interface becomes more significant, resulting in increased dispersion of the anisotropic extent of the peak shear stress. Additionally, the dispersion of the anisotropy extent increases with decreasing temperature since it increases the ice bondage in the forward shear. The variation of the anisotropy extent with normal stresses and temperatures at N = 30 is shown in Figure 7. The anisotropic extent of dispersion increases with increasing normal stress, but the difference was significantly smaller than for N = 1. After the 30th cycle, the particle distribution on the specimen’s surface is stable, and the forward and reverse friction effects are similar. Therefore, it is less affected by variations in normal stress. With decreasing temperature, the dispersion of the anisotropy extent increases. However, the anisotropy extent is not sensitive to temperature after the 30th cycle, and it varies irregularly with temperature.

The peak shear stress of cold frozen cement-treated sand shows a major reduction in the first cycle and little change after the tenth cycle. Therefore, a nonlinear logarithmic formula is used to fit the experimental data within the tenth cycle. The fitted regression equation is given as follows:*τ*_f_ = a – b × ln(N + c)(1)
where _f_ is the peak shear stress a and b are related to the first peak shear stress and the peak shear stress decay rate, respectively. The regression coefficient c is equal to −1. Figure 8 depicts the fitting curves for various temperatures. The following equations can be used to calculate the parameters a and b, which change linearly with normal stress *σ*_N_:a = *k*_1_*σ*_N_ + *k*_2_(2)
b = *k*_3_*σ*_N_ + *k*_4_(3)

Substituting Equations (2) and (3) in Equation (1), the relationship between *τ*_f_ (kPa), N, and *σ*_N_ (kPa) at −4 °C–−14 °C is obtained as follows:*τ*_f_ = (*k*_1_*σ*_N_ + *k*_2_) − (*k*_3_*σ*_N_ + *k*_4_) × ln(N − 1)(4)
where, *k*_1_, *k*_2_, *k*_3_, and *k*_4_ are the parameters shown in Table 3. It must be noted that the values of *k*_3_ and *k*_4_ are abnormal at −10 °C, indicating a different value of b than other temperatures. As seen in Figure 8, the decay rate of −10 °C is significantly lower than other temperatures, which repeated experiments have verified. Furthermore, this phenomenon, which is primarily due to the different variations of ice crystals on the interface at different temperatures and normal stresses under cyclic shear, will be discussed in subsequent chapters.

The variation of the peak shear stress in the 1st and 30th cycles at various temperatures is shown in Figure 9. The peak shear stress of cold frozen cement-treated sand is greater than that of warm frozen cement-treated sand in the first cycle. Friction and ice bondage are the primary sources of peak shear stresses in warm frozen cement-treated sand and cold frozen cement-treated sand, respectively. Additionally, the ice bondage is high in the first cycle. The dispersion of warm frozen cement-treated sand is significantly larger than that of cold frozen cement-treated sand, indicating that low-temperature change affects ice bondage significantly. However, the amount of ice crystals in the warm frozen cement-treated sand increases slightly with decreasing temperature, resulting in little effect on the peak shear stress.

In the 30th cycle, the peak shear stress of cold frozen cement-treated sand is smaller compared to warm frozen cement-treated sand. After multiple cycles, the ice bondage disappears, and the peak shear stress is due to friction and fracture of the frozen cement-treated sand at the edge of the dent. The interface becomes smooth as the ice crystals melt and re-freeze during shearing, resulting in significantly reduced friction and lower peak shear stress. The average peak shear stress of the 30th cycle for cold frozen cement-treated sand under 4 normal stresses accounts for only 26.8–37.9% of the 1st cycle. However, the difference between the peak shear stress of the 30th and 1st cycles for warm frozen cement-treated sand is 0.3–51.2 kPa, indicating that cyclic shearing does not change friction significantly.

The variation of the peak shear stress with temperature under different normal stresses is shown in Figure 10, and the data in previous publications are quoted and compared with the frozen cement-treated sand–concrete interface. Wen et al. [44] conducted direct shear tests on the adfreeze interface between silt and concrete with a moisture content of 21.5%, and Huang et al. [45] studied the silty clay soil-ice interface with a moisture content of 23%. It can be seen that the peak shear stress of the three interfaces increases with decreasing temperature when the normal stress is less than 500 kPa. This indicates that the ice bondage increases with decreasing temperature, and the peak shear stress is mainly provided by the bonding effect. However, when the normal stress is not less than 500 kPa, the peak shear stress of frozen cement-treated sand–concrete interface depicts a different variation. The peak shear stress decreases as the temperature drops from −1 °C to −2 °C. The ice crystals decrease with increasing temperature and normal stress. Therefore, the peak shear stress is mainly provided by friction at −1 °C. At high normal stress, the ice bondage almost disappears at −2 °C. The roughness of the interface is reduced as the ice crystals fill the pores, hence the friction decrease. There is no other research on the interface between frozen soil and structure when the normal stress is not less than 500 kPa and the temperature is not lower than −2 °C. Therefore, this result needs further attention in future research.

It is evident that friction significantly affects the shear behavior of the interface. Therefore, the Mohr–Coulomb shear failure criterion was selected to analyze the peak shear stress as follows:*τ*_f_ = *c*_f_ + *σ*_N_ tan(*ϕ*_f_)(5)
where, *c*_f_ and *ϕ*_f_ denote peak interface cohesion and peak interface friction angle, respectively. Table 4 lists the relationship variations between *τ*_f_ and *σ*_N_ for different conditions.

As shown in Figure 11, the values of *c*_f_ and *ϕ*_f_ at different temperatures for the 1st and 30th cycles were obtained by least-square regression analysis. In the first cycle, the cohesion of warm frozen cement-treated sand increases with decreasing temperature, indicating that the increase in ice crystals plays a critical role in ice bondage. However, in the 30th cycle, it decreases with decreasing temperature, primarily due to the reduced cement-treated sand bondage near the interface. Furthermore, the cohesion changes linearly with decreasing temperature, as follows:*c*_f_ = *η*_1_*T* + *η*_2_(6)
where, *η*_1_ and *η*_2_ are the parameters, and *T* denotes temperature. Figure 11b shows that the variation of the friction angle for warm frozen cement-treated sand from the 1st to the 30th cycle is small. It decreases and increases with decreasing temperature in the 1st and 30th cycles, respectively, primarily due to the influence of ice crystals. The friction angle changes linearly with decreasing temperature as follows:*ϕ*_f_ =*η*_3_*T* + *η*_4_(7)
where, *η*_3_ and *η*_4_ are the parameters. Substitution of Equations (6) and (7) for N = 1 and N = 30 in Equation (5) gives the relationship between *τ*_f_ (kPa) of warm frozen cement-treated sand, *T* (°C), and *σ*_N_ (kPa) as follows:*τ*_f_ = (*η*_1_*T* + *η*_2_) + *σ*_N_ tan(*η*_3_*T* + *η*_4_)(8)

Cohesion is reduced in the 30th cycle for cold frozen cement-treated sand since ice bondage disappears after the 1st cycle. Furthermore, cohesion increases significantly as the temperature decreases, and the change is linear, as given in Equation (6).

As shown in Figure 11b, the friction angle of the cold frozen cement-treated sand decreases significantly from the 1st to the 30th cycle, indicating a significant reduction in friction. The surface characteristics of the frozen cement-treated sand after testing at −10 °C are shown in Figure 12d. The ice aggregates are continuously compacted, and the surface is smoothed during the shearing process, resulting in a significantly decreased friction angle. In the first cycle, the friction angle increases with decreasing temperature, indicating that increasing ice crystals increases the friction force. However, the friction angle decreases with decreasing temperature in the 30th cycle, primarily due to the crushing and melting of more ice crystals. The friction angle changes linearly with decreasing temperature, as in Equation (7).

The relationship between *τ*_f_ (kPa) of cold frozen cement-treated sand, *T* (°C), and *σ*_N_ (kPa) is the same as Equation (8). The parameters *η*_1_, *η*_2_, *η*_3_, and *η*_4_ are shown in Table 5.

Note that the friction angle decreases as the temperature drops from −1 °C to −2 °C; the same conclusion has been reached in the studies of the frozen silty soil–structure interface [45,46]. The ice crystals fill the pores in the soil near the interface, and the roughness of the interface is reduced by the formation of ice aggregates. As the temperature decreases, the ice crystals increase and the roughness decreases gradually. The image of the specimen surface after testing also confirmed the existence of ice aggregates on the interface (Figure 11). On the contrary, Huang et al. [45] and Shi et al. [47] studied the ice-soil interface, and they found that the friction angle increases with decreasing temperature. The formation mechanism of the friction angle of the two interfaces is different. When the interface of the soil–structure is frozen, the soil and ice crystals are cemented to form friction with the structure, and the structure is not affected by temperature. However, the soil and ice at the ice-soil interface produce friction, Ice crystals increase with decreasing temperature. Therefore, the adhesion between soil and ice increases, and the friction angle increases.

The surface images of cement-treated sand specimens at different temperatures before and after testing are shown in Figure 12. The bondage effect is not strong during initial shearing since the original state of the cement-treated sand surface has distinct particles and is in direct contact with the shear plate. At 25 °C, there are fewer individual particles and more aggregates, and the shearing process repeatedly squeezes and breaks the cement-treated sand. The continuous fusion of particles and aggregates enhances the bonding force. Additionally, enough cement-treated sand is attached to the shear plate at this time, which forms a certain bonding effect with the cement-treated sand on the surface of the specimen. It increases the cohesion from 6.95 kPa to 33.85 kPa during testing. The cohesion of the 30th cycle at −1 °C and −1.5 °C is higher than at 25 °C. As shown in Figure 12c, ice crystals formed on the surface are pressed and melted during the shearing process, and then they re-bond with the particles and aggregates, further enhancing the bonding effect. Ice creep has a significant impact on the shear behavior of frozen sand. At −1 °C, there are a few ice crystals on the surface of the specimen, as shown in Figure 12, and the ice bondage formed in the interface is small. Because the friction is greater than the ice bondage, and the amount of ice melting is high during the shearing process. Therefore, because the temperature is close to the melting point, the peak shear stress is primarily due to friction, and the change is insignificant. At −10 °C, ice and soil particles form stable aggregates and form a strong bond at the interface. Therefore, the peak shear stress is high in the first few cycles. After the peak shear stress of the first cycle, the ice is brittle and fractured, resulting in a significantly lower peak shear stress. He et al. [48] found the frozen soil–concrete interface exhibits a strain-hardening behavior at −1 °C, while it exhibits strain-softening behavior at −5 °C. They attributed it to the differences in the number of ice crystals on the surface, which caused the differences in creep behavior during the shearing process. This is consistent with the conclusion of this paper, indicating that ice crystals play a vital role in different frozen interfaces.

### 3.3. Variation of the Peak Normal Displacement with the Number of Cycles at Various Conditions

The development of the peak normal displacement with the number of cycles at various temperatures and normal stresses is shown in Figure 13. The peak normal displacement increases, but the increase rate gradually decreases with the increasing number of cycles. However, the increase in the rate of peak normal displacement increases with increasing normal stress. The peak normal displacement of warm frozen cement-treated sand is significantly larger than that of cold frozen cement-treated sand. The warm frozen cement-treated sand increases continuously, while the cold frozen cement-treated sand increases very little with the increasing number of cycles, resulting in a larger difference. Furthermore, this can be attributed to the formation of fewer ice crystals at the interface when the temperature is higher than −2 °C and the effect of ice bondage is insignificant. The frozen cement-treated sand is continuously broken and melted during testing. The strength of the specimen is low at higher temperatures, resulting in a continuous increase in peak normal displacement. At 25 °C, the peak normal displacement gradually increases, but the peak shear stress is basically unchanged (Figure 5a). Zhang et al. [49] obtained the same results when studying the gravel-structure interface and attributed it to persistent particle fragmentation. However, the shear stress of warm frozen cement-treated sand changes significantly in the first few cycles (Figure 5b–d). This indicates that the ice crystals affect the bondage of the interface and have an effect on the shear stress. When the temperature is lower than −2 °C, the ice bondage is significantly increased by a large number of ice crystals, and little breakage occurs during testing. Especially for low temperature and small normal stress, the surface of the specimen is continuously compacted, and no new breakage occurs, resulting in a small increase in the peak normal displacement.

Figure 14a depicts the variation of the maximum normal displacement with temperature for various normal stresses. Because of significant ice bondage at lower temperatures, the maximum normal displacement under the same normal stress decreases with decreasing temperature, resulting in less breakage. Furthermore, for higher normal stresses, the maximum normal displacement decreases faster. The increase in normal stress presses and melts the ice crystals at the surface faster, resulting in increased breakage that leads to the temperature significantly impacting the variation of the normal displacement. The variation of the maximum normal displacement with the normal stress at different temperatures is plotted, as shown in Figure 14b. The maximum normal displacement of the warm frozen cement-treated sand is significantly higher than that of the cold frozen cement-treated sand, indicating that the ice crystals formed on the surface and the strength of the frozen cement-treated sand have significant differences at different temperatures.

For the same temperature, the maximum normal displacement increased with increasing normal stress, and this is consistent with the results of Zhao et al. [35] and Chang et al. [50]. However, the result is different from the research on the unfrozen sand-structure interface conducted by Pra-ai et al. [51]. Their study showed that the maximum normal displacement decreased with increasing normal stress. Chang et al. [50] attributed it to the pressure-melting behavior of frozen soil, which led to a decrease in the strength of the frozen soil. The results of this study verify this mechanism as follows: there are few ice crystals on the surface of warm frozen cement-treated sand, and the strength decreases significantly as the ice crystals are continuously broken and melted. Therefore, the normal displacement increases significantly with increasing normal stress.

The relationship between the maximum normal displacement of frozen cement-treated sand and temperature under four normal stresses is fitted. The fitted regression equation is given as follows:*ν*_max =_ m|T|^n^(9)
where, *ν*_max_ denotes the maximum normal displacement. m and n are the parameters. As shown in Figure 15, the measured values are in good agreement with the exponential Equation (9).

The parameters m and n change linearly with *σ*_N_, and the obtained relationship between them is as follows:m = 0.0037*σ*_N_ − 0.018(10)
n = −0.0005*σ*_N_ − 0.765(11)

Substitution of Equations (10) and (11) in Equation (9) gives the relationship between *ν*_max_ (mm) of frozen cement-treated sand, *T* (°C), and *σ*_N_ (kPa) as follows:*ν*_max_ = (0.0037*σ*_N_ − 0.018)|*T*|^(−0.0005^*^σ^*^N − 0.765)^(12)

### 3.4. Variation of the Shear Stiffness with the Number of Cycles at Various Conditions

The cyclic shear behavior of the frozen cement-treated sand–concrete interface is analyzed using the shear stiffness *K* of the soil, which is calculated as follows:(13)$$K=\frac{K_{1}+K_{2}}{2}=\frac{\tau_{1}+\tau_{2}}{2\Delta a}$$
where, *K*_1_ and *K*_2_ and *τ*_1_ and *τ*_2_ denote shear stiffnesses and maximum shear stresses in the two shear directions, respectively. Δ*a* denotes the displacement semi-amplitude.

Figure 16 depicts the variation of shear stiffness with cycle number under various conditions. Shear stiffness increases with increasing normal stress at different temperatures. The shear stiffness changes slightly with the number of cycles at 25 °C, indicating that there is no noticeable shear hardening or shear softening during testing. The cement-treated sand interface characteristics do not change under cyclic shear at positive temperatures, and the deformation resistance remains unchanged. At −1–−4 °C, several different trends of the shear stiffness are observed with the increasing number of cycles, which are attributed to the different variations of ice crystals on the surface during testing at various conditions, resulting in different trends of deformation of frozen cement-treated sand under specific temperature and normal stress. For example, at −1 °C and 700 kPa, ice crystals on the surface are gradually melted and are difficult to re-freeze under high normal stress. In the beginning, the friction is provided by the ice crystals and cement-treated sand, and then gradually, it is only provided by the cement-treated sand, exhibiting a strain-softening behavior. At −1 °C and 100 kPa, the melting of ice crystals is less, and the surface of the cement-treated sand is continuously broken, resulting in increased friction and a strain-hardening behavior. The interface has a strong deformation resistance at −4 °C and 700 kPa due to the ice bondage in the first cycle. However, the shear stiffness decreases sharply as the ice bondage disappears, and then the continuous breaking of the frozen cement-treated sand under high normal stress gradually increases the stiffness. At −6 °C to −14 °C, the shear stiffness decreases with the increasing number of cycles and exhibits a strain-softening behavior. At the beginning of cyclic shear, the attenuation is remarkable, but the shear stiffness decreases and tends to stabilize, indicating a constant weakening of the deformation resistance of the interface. The ice crystals on the surface are continuously melted and re-frozen with the cement-treated sand during testing, resulting in a smoother interface showing a small shear stiffness.

The variation of the shear stiffness with the number of cycles at various normal stresses is shown in Figure 17. The shear stiffness of the first cycle first decreases and then increases with decreasing temperature. The minimum value of the shear stiffness appears at −4 °C due to the worst coupling effects of ice bondage and friction. When the temperature is higher or lower than −4 °C, then friction and ice bondage play the leading roles, respectively. The shear stiffness of the 30th cycle first increases and then decreases. When the normal stresses are 100 kPa and 300~700 kPa, the maximum values appear at −1 °C and −2 °C, respectively, and the shear stiffness decreases significantly after reaching the maximum value. The shear stiffness of the warm frozen cement-treated sand changes slightly from the 1st to the 30th cycle, while it changes considerably for the cold frozen cement-treated sand, which can be attributed to the significant ice bondage in the 1st cycle that increases the shear stiffness significantly. The shear stiffness of the 1st cycle was larger compared to the 30th cycle, except for −4 °C and 700 kPa. The shear stiffness of the 30th cycle is primarily due to the continuous breakage on the surface under high normal stress, which enhances deformation resistance.

### 3.5. Variation of the Surface and Peak Shear Stress of Frozen Cement-Treated Sand with Different Number of Cycles at T = −10 °C, σ_N_ = 300 kPa

In this study, the shear stress gradually decreases from its peak under certain conditions. A similar phenomenon has been reported previously [33], attributed to the tumbling and breaking of soil particles and ice crystals. However, no experimental proof was given for this phenomenon. Therefore, the most noticeable attenuation phenomenon at −10 °C and 300 kPa was selected to perform the direct shear tests under N = 1, 5, 10, 20, and 30 to verify this mechanism. Figure 18 shows surface images revealing the interface conditions after testing for the set number of cycles. After testing, sharpened brightness adjustment and binarization were performed to obtain a contrasting surface image for easier observation of surface morphology. Figure 18b depicts the processed image. The black zone is where ice crystals and soil particles are smoothed, while the white zone is where they are not. As shown in Figure 19a, the rough area has many ice crystals, while the smooth area has strong integrity. The crushed and broken ice crystals are re-bonded with cement-treated sand to form the aggregates. As shown in the mesoscopic image of the smooth area in Figure 19b, it has strong integrity, and the melted and crushed ice crystals re-bond with the cement-treated sand to form smooth aggregates. As shown in the mesoscopic image of the rough area in Figure 19c, it has many unbroken ice crystals and cement-treated sand particles.

The black zone increases with the increasing number of cycles, indicating an increase in the area of ice crystals and soil particles being smoothed. The surface becomes smooth, and the friction angle and cohesion are gradually reduced, resulting in decreased shear stress. When exposed to high temperatures or large normal stresses, the ice crystals on the surface melt quickly, and soil particle deformation reaches the stable phase within a few cycles. For low temperatures or small normal stresses, the deformation of the ice crystals and soil particles is not noticeable, reaching the stable phase quickly.

The black zone, the area where ice crystals and soil particles are smoothed, is counted by image processing software Image-Pro Plus 6.0. Its proportion of the entire image is defined as smoothness *D*, which is given as follows:(14)$$D=100\times \frac{S_{1}}{S}$$
where *S*_1_ and *S* denote the area of the black zone and the total area, respectively. The smoothness after various cycles is measured and compared to the shear stress during the same cycle. As shown in Figure 20, the smoothness increases linearly, and the shear stress decreases exponentially with the increasing number of cycles, indicating that the dominant reasons for reduced shear stress are the melting and breaking of ice crystals and the deformation of soil particles during testing. The fitting formulae for *D* and shear stress *τ* with the number of cycles are obtained as follows:*D* = 0.9632N + 37.242(15)
*τ* = 286.11N^−0.457^(16)

Substituting Equation (15) into Equation (16), the obtained relationship between *τ* and *D* is as follows:(17)$$\tau =286.11{\left(\frac{D-37.242}{0.9632}\right)}^{-0.457}$$

It must be noted that Equation (17) is valid only under the test conditions of this study, and the methods of constructing formulae under other test conditions require further experimental research.

## 4. Discussion

This study investigates the cyclic shear behavior of the frozen cement-treated sand–concrete interface. The test conditions in this study are the same as those in a previously reported systematic study of the cyclic shear behavior of the frozen sand–concrete interface [35]. Therefore, the data on the variation of peak shear stress and peak normal displacement of the frozen sand-structure interface at −2 °C and −14 °C are quoted and compared with that of frozen cement-treated sand. As shown in Figure 21, ‘S’ is the abbreviation for ‘sand’, and ‘CS’ is the abbreviation for ‘cement-treated sand’. The peak shear stress of the first cycles of frozen sand and frozen cement-treated sand increases with increasing normal stress and decreasing temperature. The cyclic shear behavior of frozen sand is the same at −2 °C and −14 °C, showing a peak in the first cycle and then a significant decrease. The behavior of frozen cement-treated sand followed the same trend at −14 °C. However, at −2 °C, the shear behavior shows several types of variation due to the different trends of ice crystals at various normal stresses. Consequently, the peak shear stress of the interface in the first cycle is greater than that of the frozen cement-treated sand at the two temperatures. However, the peak shear stress of the frozen cement-treated sand–concrete interface approaches or exceeds that of the frozen sand with an increasing number of cycles at −2 °C. Furthermore, this is because the shear stress is mainly generated by the sliding friction after the disappearance of the ice bondage, when the shear stress of the interface peaks. The frozen sand surface is stabilized as more ice crystals are formed, and the peak shear stress of each cycle remains unchanged. However, at −2 °C, the strength of frozen cemented sand is low, and ice crystals are melted during the test, resulting in severely broken cemented sand and increased friction, gradually increasing the peak shear stress. At −14 °C, the peak shear stress of the frozen sand-structure interface during the test is greater than that of the frozen cement-treated sand, which is due to more ice crystals at the interface forming a notable ice bond.

The differences in peak normal displacement between frozen cement-treated sand–concrete and frozen sand-structure interfaces are shown in Figure 22. The peak normal displacement increases with the increasing number of cycles. At the same temperature, the peak normal displacement increased with increasing normal stress. However, the peak normal displacement decreases with decreasing temperature under the same normal stress. At −2 °C, the peak normal displacement of the frozen cement-treated sand–concrete interface was larger than the frozen sand under the same conditions, which is attributed to the formation of fewer ice crystals on the surface and the low strength of the frozen cement-treated sand. The frozen sand has remarkable strength at −2 °C and cannot be damaged easily. However, the peak normal displacement of frozen cement-treated sand is smaller than frozen sand at −14 °C under the same normal stress, indicating the significantly increased strength of the frozen cement-treated sand at low temperatures and the occurrence of little breakage during testing.

A significant difference is observed between the cyclic shear behavior of frozen cement-treated sand–concrete and frozen sand–concrete interfaces. When the tunnel boring machine (TBM) penetrates the artificially frozen- and cement-stabilized zones, the interface shear resistance is significantly smaller than the original soil at low temperatures. However, at −2 °C it is close, which requires timely adjustment of the driving parameters of the shield machine. This study analyzed two types of variation of the interface with the number of cycles, and the adjustments of the driving parameters should also be based on the temperature of the frozen cement-treated sand. In addition, it was found that −4 °C is the critical point of the two types of variation, indicating that if the temperature of the frozen cement-treated sand is above −4 °C, the interface under the cyclic shear of the TBM can slowly melt, resulting in severe damage from water and sand splashes. Therefore, when the TBM enters the frozen cemented sand zones, it is necessary to ensure that the interface temperature is lower than −6 °C in order to maintain the stability of the frozen cement-treated sand.

## 5. Conclusions

The cyclic shear behavior of the frozen cement-treated sand–concrete interface has been investigated in this study. The behaviors of the interface, based on the systematical experimental observations and analyses, are as follows:

(1) Variations of two types with the number of cycles are seen for the peak shear stress of warm frozen cement-treated sand and cold frozen cement-treated sand. −4 °C is the critical point of the cyclic shear behavior of the frozen cement-treated sand–concrete interface, as the normal stress does not exceed 700 kPa.

(2) The maximum normal displacement of warm frozen cement-treated sand is significantly higher than that of cold frozen cement-treated sand. The former increases continuously, while the latter increases very little with the increasing number of cycles, resulting in a larger difference.

(3) The shear stiffness of the first cycle first decreases and then increases with decreasing temperature. However, it first increases and then decreases for the 30th cycle, which is attributed to the different variations of ice crystals on the surface during testing in various conditions.

(4) The close-up images taken after testing reveal the surface conditions. They demonstrate that the melting and breaking of ice crystals and the deformation of soil particles during testing are the dominant reasons for the reduced shear stress under certain conditions.

(5) The variation of the peak shear stress and the peak normal displacement during cyclic shear have a significant difference for the frozen cement-treated sand–concrete and frozen sand–concrete interfaces, which is due to the different number of ice crystals at the interface, affecting the strength of the ice bondage.

## Figures and Tables

**Figure 1 materials-15-08756-f001:**
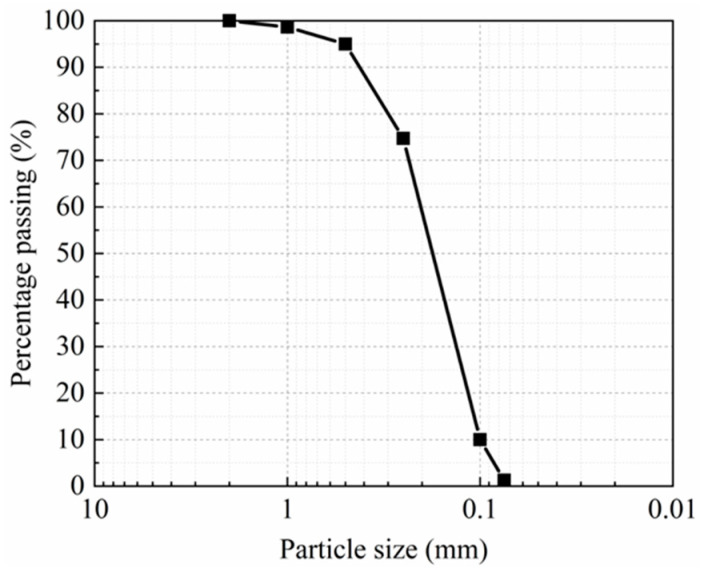
Grain-size distribution of the test soil.

**Figure 2 materials-15-08756-f002:**
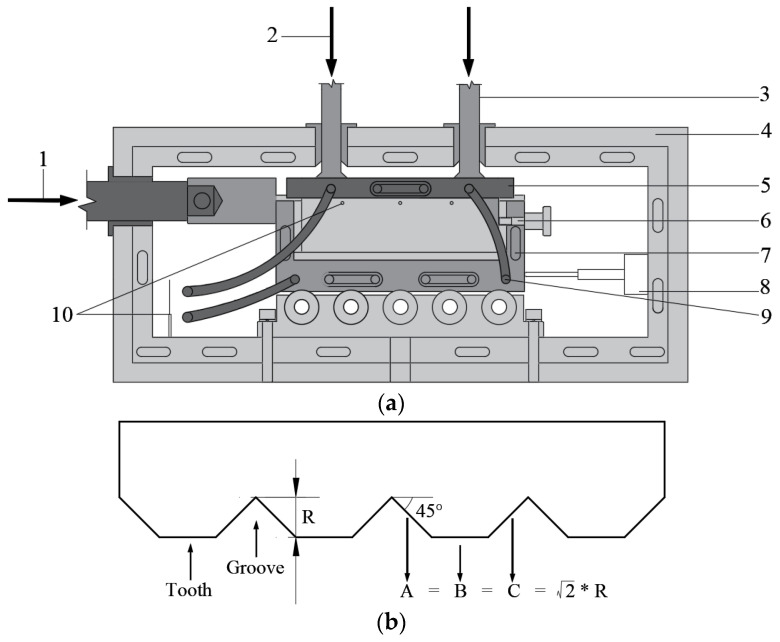
Schematic diagrams of the DDJ-1G. (**a**) Loading systems in the DDJ-1G; (**b**) schematic diagram of the shear plate surface, showing (1) horizontal loads; (2) vertical loads; (3) structural panel; (4) environmental chamber; (5) shear plate; (6) inner box; (7) shear box; (8) displacement transducers; (9) main circulation channels to supply refrigerated coolant; (10) temperature sensor.

**Figure 3 materials-15-08756-f003:**
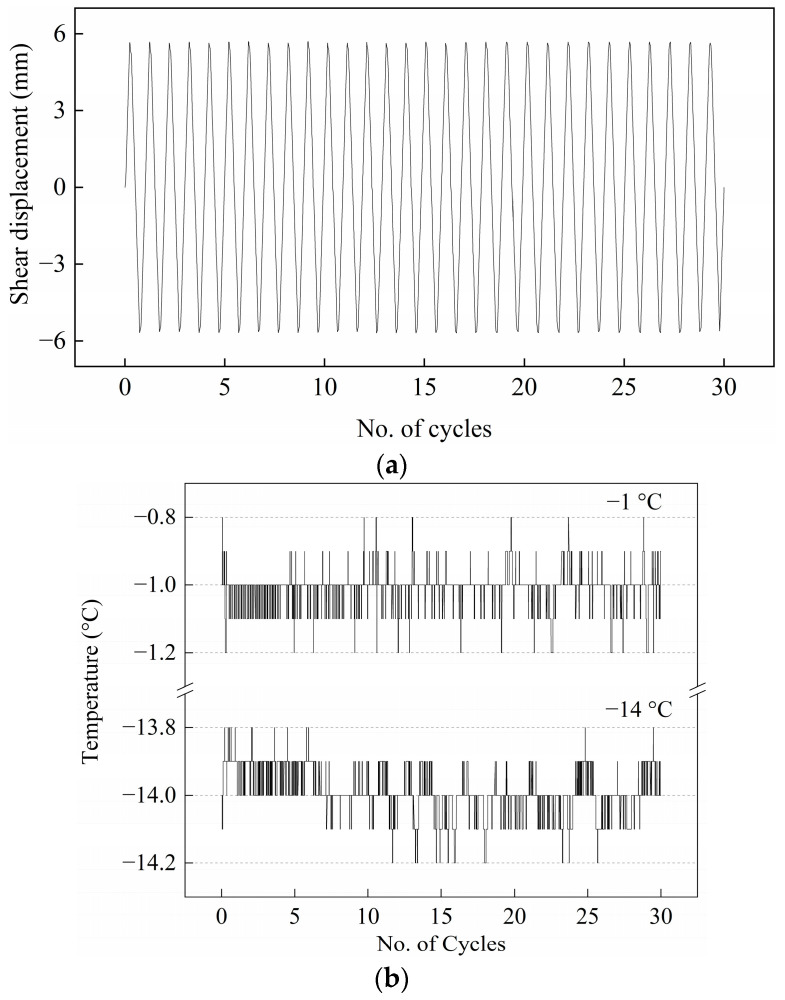
Control validation and accuracy. (**a**) Shear displacement with cyclic loading numbers in direct shear tests; (**b**) frozen cemented soil temperatures measured in constant frozen temperature tests.

**Figure 4 materials-15-08756-f004:**
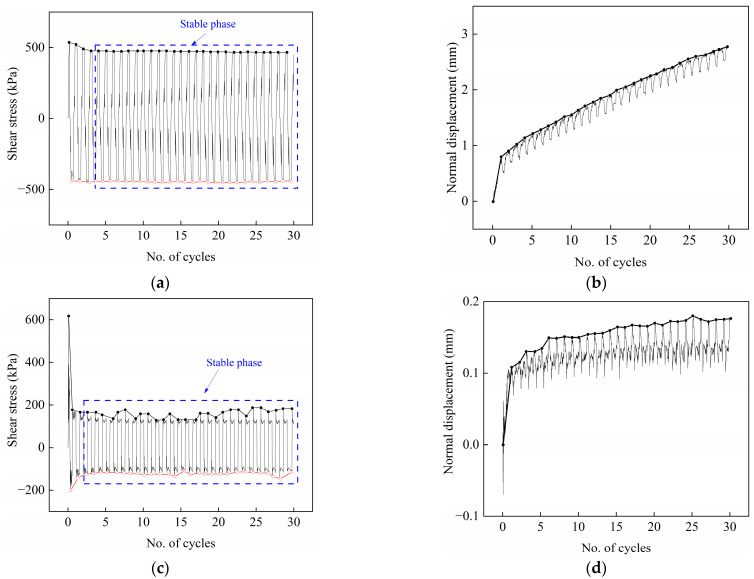
Typical variation of the shear stress and normal displacement with the number of cycles for frozen cement-treated sand–concrete interface. (**a**) Shear stress at −1 °C, 700 kPa; (**b**) normal displacement at −1 °C, 700 kPa; (**c**) shear stress at −14 °C, 700 kPa; (**d**) normal displacement at −14 °C, 700 kPa.

**Figure 5 materials-15-08756-f005:**
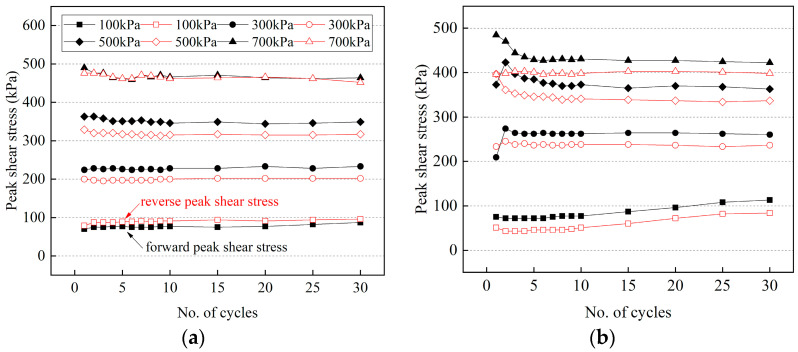
Peak shear stress with the number of cycles for cement-treated sand–concrete interface in various conditions. (**a**) 25 °C; (**b**) −1 °C; (**c**) −1.5 °C; (**d**) −2 °C; (**e**) −4 °C; (**f**) −6 °C; (**g**) −10 °C; (**h**) −14 °C.

**Figure 6 materials-15-08756-f006:**
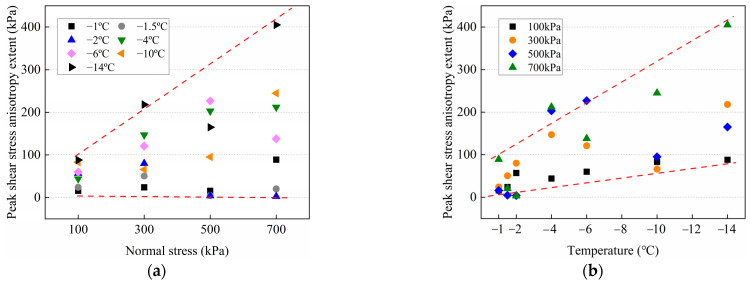
Variation of anisotropy extent of peak shear stress at N = 1. (**a**) With normal stress; (**b**) with temperature.

**Figure 7 materials-15-08756-f007:**
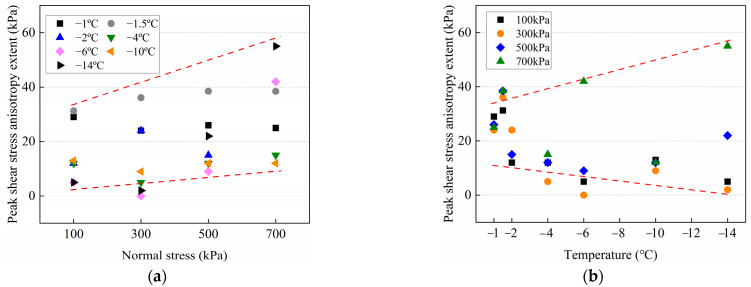
Variation of anisotropy extent of peak shear stress at N = 30. (**a**) With normal stress; (**b**) with temperature.

**Figure 8 materials-15-08756-f008:**
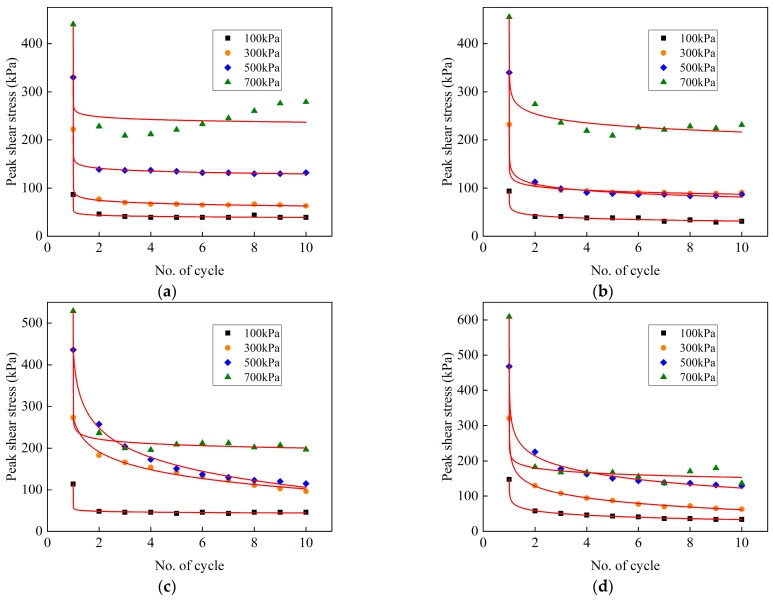
Fitting curves of the peak shear stress with the number of cycles under different normal stresses at −4 °C–−14 °C. (**a**) −4 °C; (**b**) −6 °C; (**c**) −10 °C; (**d**) −14 °C.

**Figure 9 materials-15-08756-f009:**
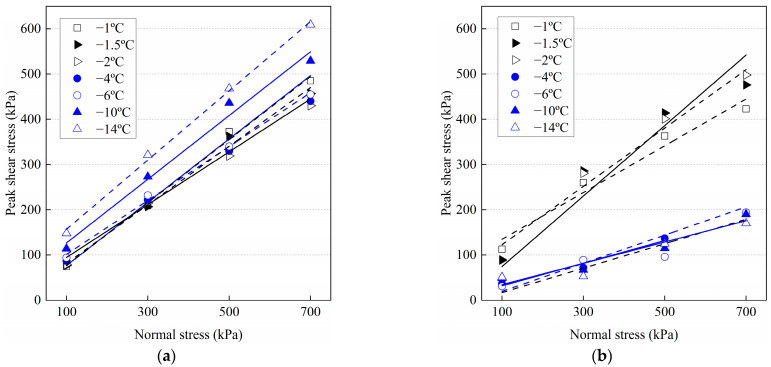
Variation of the peak shear stress with the different number of cycles at various temperatures. (**a**) N = 1; (**b**) N = 30.

**Figure 10 materials-15-08756-f010:**
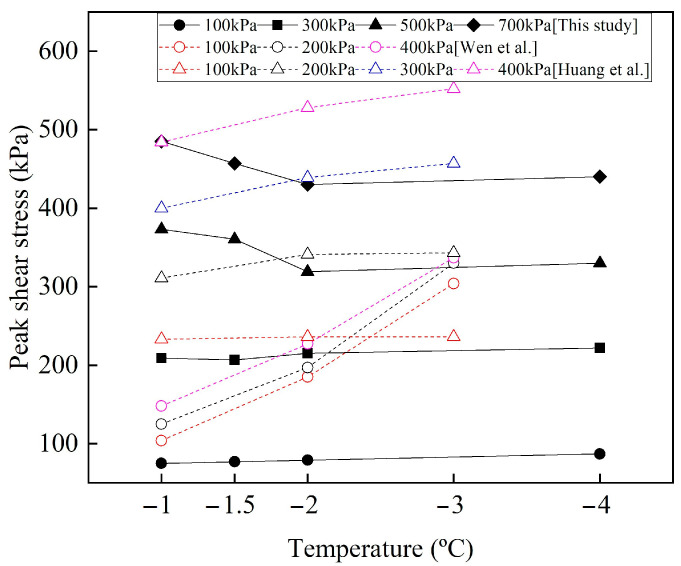
Variation of the peak shear stress with temperature for N = 1 and compared with previous publications [44,45].

**Figure 11 materials-15-08756-f011:**
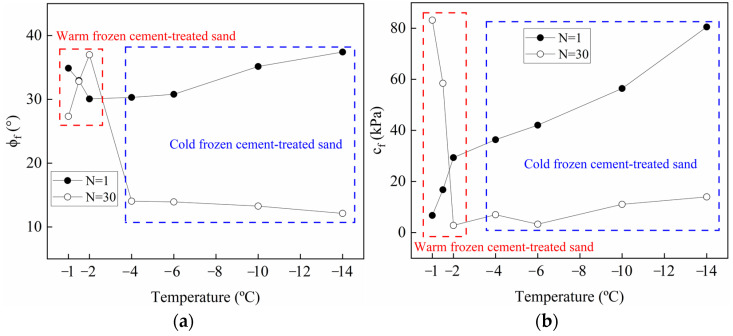
Variation of friction angle and cohesion for N = 1 and N = 30 at different temperatures. (**a**) Cohesion; (**b**) friction angle.

**Figure 12 materials-15-08756-f012:**
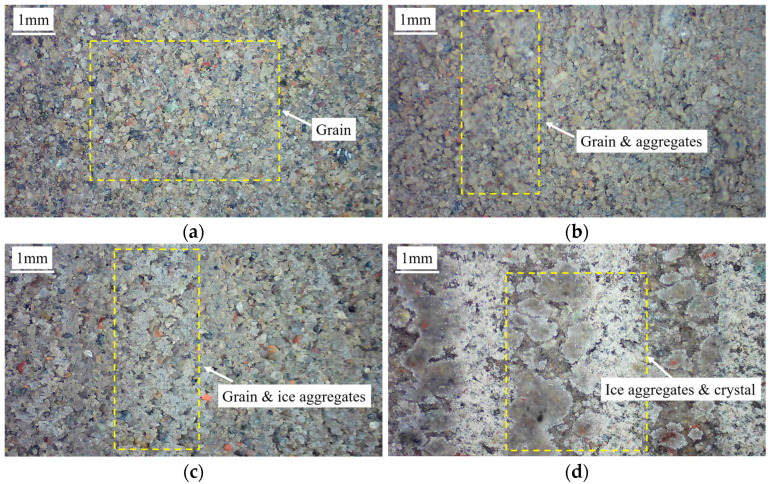
Surface images of the cement-treated sand specimens before and after testing. (**a**) Before testing; (**b**) after testing at 25 °C; (**c**) after testing at −1 °C; (**d**) after testing at −10 °C.

**Figure 13 materials-15-08756-f013:**
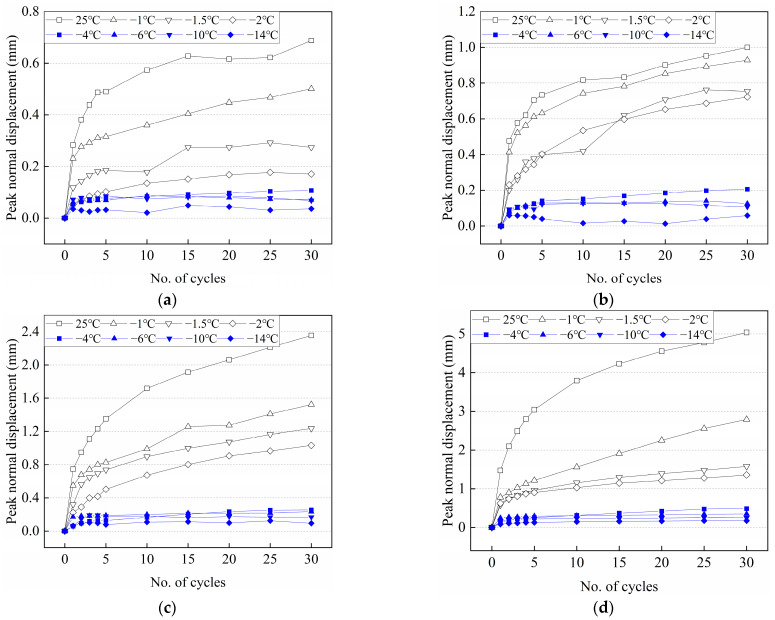
Peak normal displacement with the number of cycles for cement-treated sand–concrete interface at various conditions. (**a**) 100 kPa; (**b**) 300 kPa; (**c**) 500 kPa; (**d**) 700 kPa.

**Figure 14 materials-15-08756-f014:**
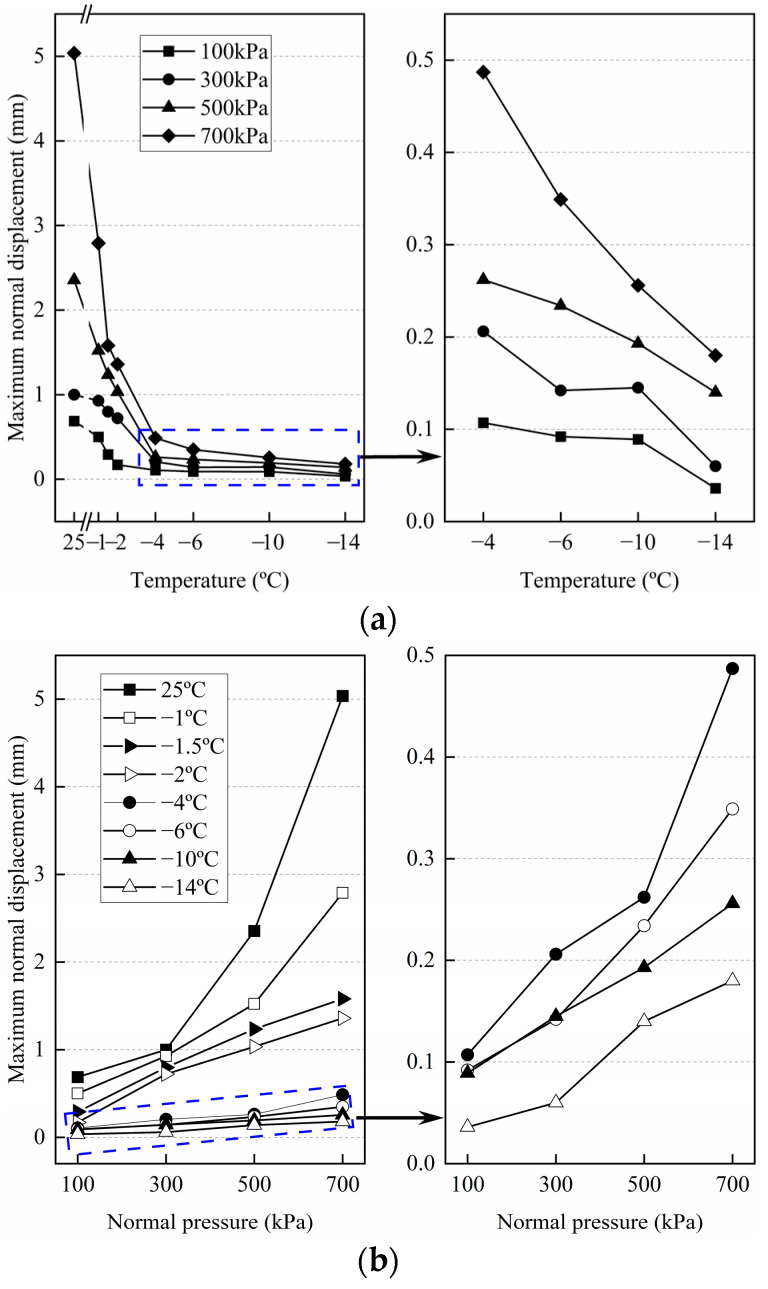
Variation of the maximum normal displacement in various conditions. (**a**) with temperature; (**b**) with normal stress.

**Figure 15 materials-15-08756-f015:**
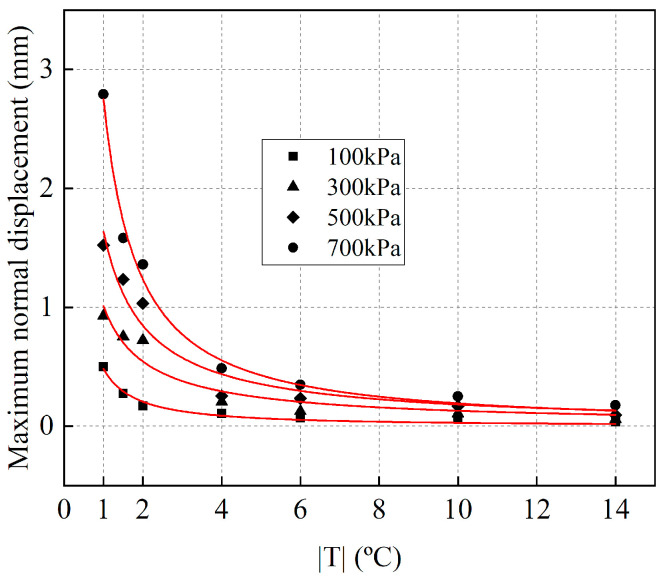
Fitting curve of the maximum normal displacements with temperatures for different normal stresses.

**Figure 16 materials-15-08756-f016:**
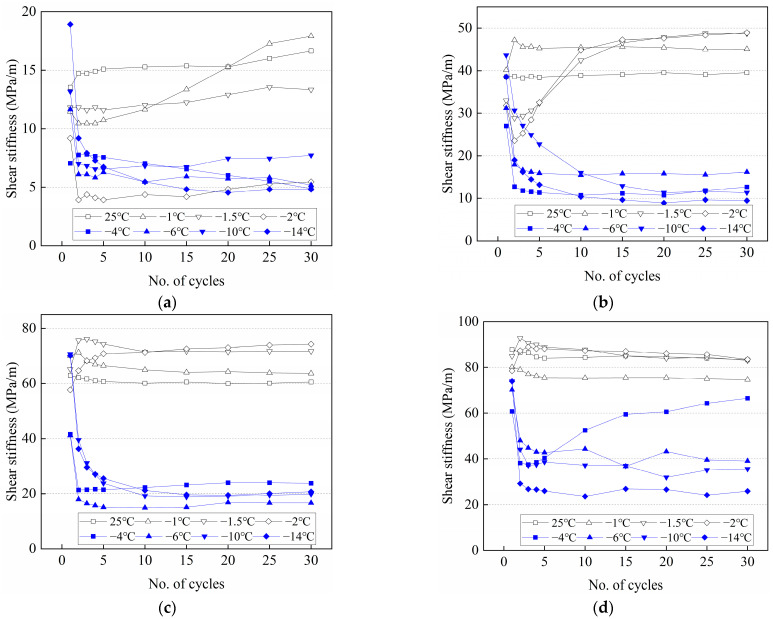
Variation of the shear stiffness with the number of cycles in various conditions. (**a**) 100 kPa; (**b**) 300 kPa; (**c**) 500 kPa; (**d**) 700 kPa.

**Figure 17 materials-15-08756-f017:**
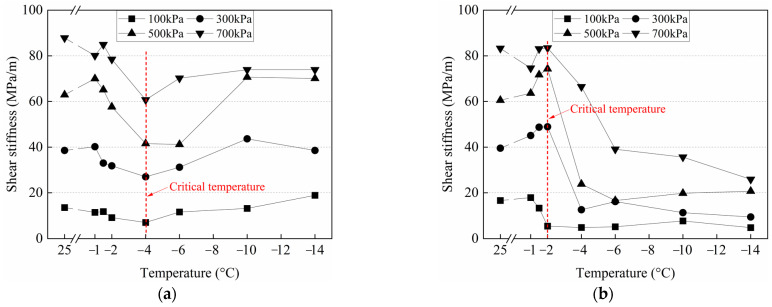
Variation of the shear stiffness with the number of cycles at various normal stresses. (**a**) N = 1; (**b**) N = 30.

**Figure 18 materials-15-08756-f018:**
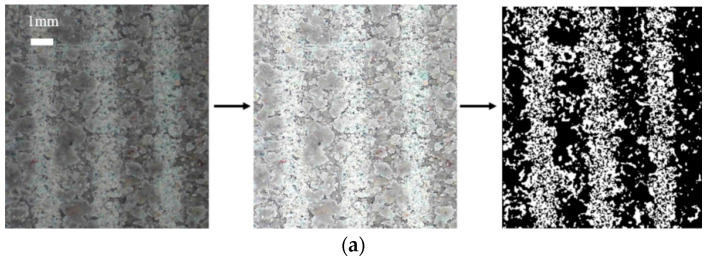
Surface image enhancement for frozen cement-treated sand specimens after testing at the different number of cycles. (**a**) Example of surface image enhancement for a frozen cement-treated sand specimen after testing; (**b**) Surface image enhancement for different number of cycles with *T =* −10 °C, *σ*_N_ = 300 kPa.

**Figure 19 materials-15-08756-f019:**
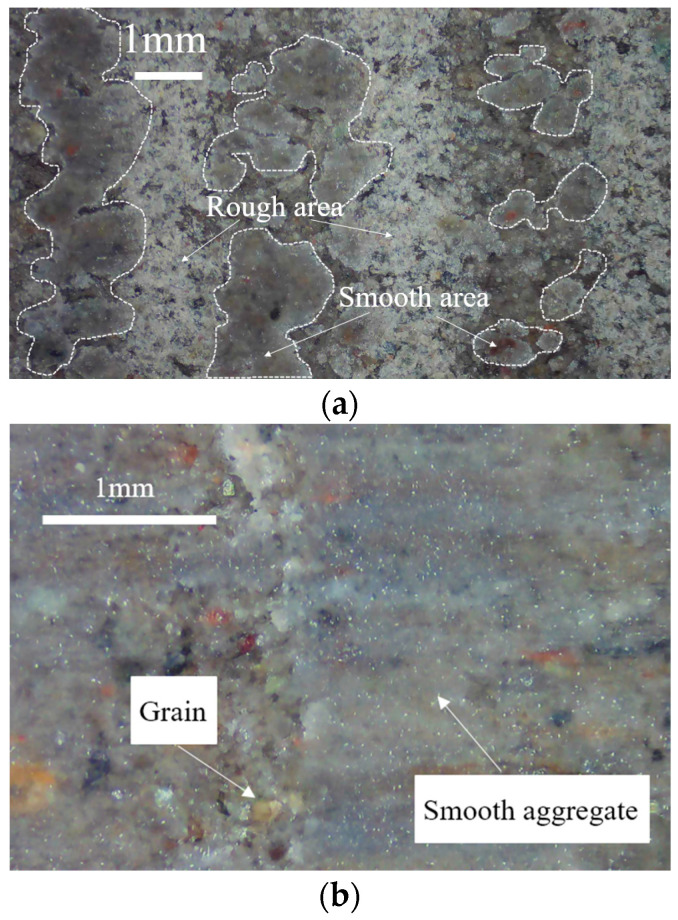
Surface image of the frozen cement-treated sand specimen after testing. (**a**) Division of the surface; (**b**) smooth area; (**c**) rough area.

**Figure 20 materials-15-08756-f020:**
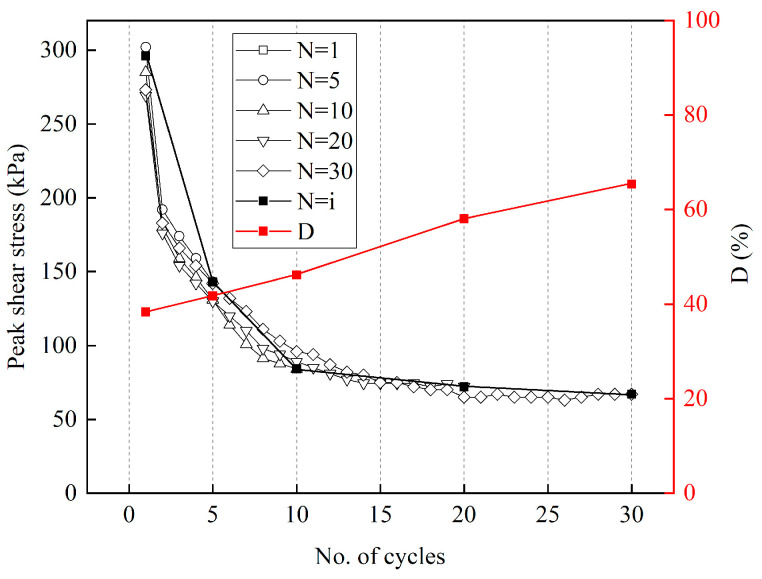
Variation of the peak shear stress and smoothness for the different number of cycles.

**Figure 21 materials-15-08756-f021:**
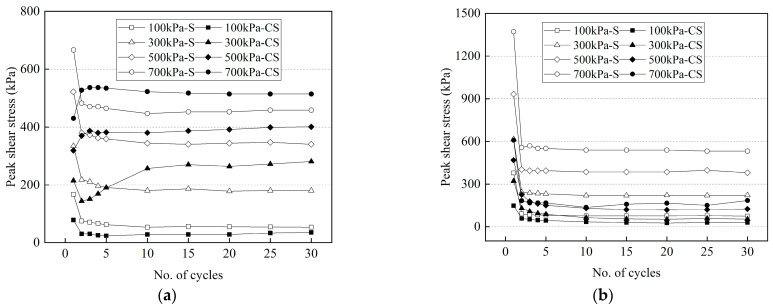
Differences of peak shear stress between frozen cement-treated sand–concrete and frozen sand–concrete interfaces. (**a**) −2 °C; (**b**) −14 °C.

**Figure 22 materials-15-08756-f022:**
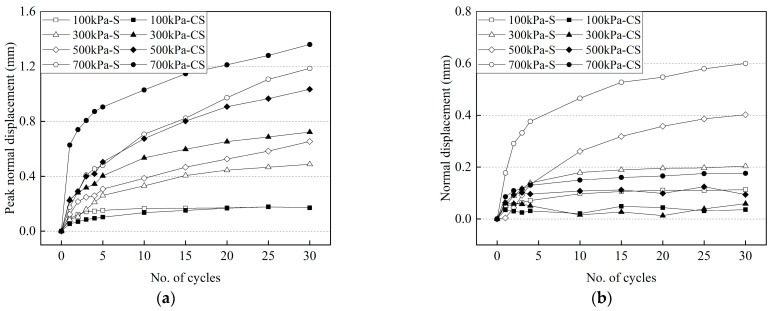
Differences of normal displacement between frozen cement-treated sand–concrete and frozen sand–concrete interfaces. (**a**) −2 °C; (**b**) −14 °C.

**Table 1 materials-15-08756-t001:** Parameters of the test soil.

Specific Gravity	Saturated Unit Weight	Natural Water Content	Void Ratio	Internal Friction Angle
	/(kN/m^3^)	/%		/Degree
2.67	19.3	27	0.721	30.2

**Table 2 materials-15-08756-t002:** Testing parameters.

Normal Stress/kPa	Testing Temperature/°C
100	25, −1, −1.5, −2, −4, −6, −10, −14
300	25, −1, −1.5, −2, −4, −6, −10, −14
500	25, −1, −1.5, −2, −4, −6, −10, −14
700	25, −1, −1.5, −2, −4, −6, −10, −14

**Table 3 materials-15-08756-t003:** Parameters *k*_1_, *k*_2_, *k*_3_, and *k*_4_ under different temperatures.

Temperature/°C	*k* _1_	*k* _2_	*k* _3_	*k* _4_
−4	0.3119	−1.9415	0.0063	2.172
−6	0.3184	0.1785	0.0199	2.6085
−10	0.5104	12.872	0.1623	−11.553
−14	0.3905	16.933	0.0752	5.4467

**Table 4 materials-15-08756-t004:** Variation of the relationship between *τ*_f_ and *σ*_N_ for N = 1 and N = 30 at different temperatures.

Temperature/°C		N = 1		N = 30	
		Relationship	R^2^	Relationship	R^2^
Warm frozen cement-treated sand	−1	*τ*_f_ = 6.7 + 0.697*σ*_N_	0.9953	*τ*_f_ = 83.15 + 0.517*σ*_N_	0.9658
	−1.5	*τ*_f_ = 16.75 + 0.648*σ*_N_	0.9921	*τ*_f_ = 58.45 + 0.645*σ*_N_	0.9480
	−2	*τ*_f_ = 29.35 + 0.579*σ*_N_	0.9965	*τ*_f_ = 2.8 + 0.753*σ*_N_	0.9498
Cold frozen cement-treated sand	−4	*τ*_f_ = 36.35 + 0.584*σ*_N_	0.9956	*τ*_f_ = 7 + 0.25*σ*_N_	0.9418
	−6	*τ*_f_ = 42.05 + 0.586*σ*_N_	0.9972	*τ*_f_ = 3.3 + 0.248*σ*_N_	0.8973
	−10	*τ*_f_ = 56.4 + 0.704*σ*_N_	0.9864	*τ*_f_ = 11.05 + 0.236*σ*_N_	0.9317
	−14	*τ*_f_ = 80.5 + 0.765*σ*_N_	0.9977	*τ*_f_ = 13.95 + 0.215*σ*_N_	0.9080

**Table 5 materials-15-08756-t005:** Parameters *η*_1_, *η*_2_, *η*_3_, and *η*_4_ under different temperatures and the number of cycles.

Temperature/°C	N	*η* _1_	*η* _2_	*η* _3_	*η* _4_
Warm frozen cement-treated sand	1	−22.65	16.38	4.81	39.85
	30	80.35	168.66	9.64	17.92
Cold frozen cement-treated sand	1	−4.401	15.92	−0.77	26.9
	30	−0.91	1.11	0.19	14.99

## Data Availability

Not applicable.

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
