# Peer review of "Cyclic Shear Behavior of Frozen Cement-Treated Sand–Concrete Interface"

_materials, 2022, doi:10.3390/ma15248756_

Round 1

Reviewer 1 Report

1. Lines 67-73: The authors here that the results are affected by the size of the specimen and provide some references to works done previously. However, the authors only list and mention what kind of experiments were done but do not mention anything about the results obtained in these works. I suggest that the authors briefly outline the main conclusions/observations in those works.

2. Figure 5: Firstly, the legend in a and b is cut off (the units for 300 and 500kPa). I suggest that the authors fix this issue. Secondly, what is the difference between red and black color? It is not stated anywhere in the figure or caption. The authors have to clearly state and indicate what is what.

3. This comment is about behavior at different temperatures shown in Figure 5. There is a clear difference in behavior between Fig.5 b,c,d and Fif 5 e,f,g,h at the very beginning of cycling, i.e. the first few cycles. As temperature drops, there is a clear drop in peak shear stress. Creep in ice is very temperature dependent, and there is a huge difference between temperatures close to the melting point (-1to-3C) and temperatures let's say below -5 or below -10C. Therefore, I am wondering whether differences in creep in ice can have any effect on the behavior of the ice-rich frozen sand at different temperatures.

4. Lines 492-493: Please fix the reference. 

Author Response

The authors appreciate the comments from reviewer 1. The questions and comments are very encouraging, constructive, and helpful in improving the quality of this manuscript. Detailed response to each question is presented below. The additions/revisions in the manuscript are highlighted in red.

Reviewer 2 Report

General comments

Please, check the language. There are many unintelligible sentences.

The expression “warm and ice-rich frozen cemented soil and structure interface” is strange. I suggest other designation…

The expression “cemented sand” is used, which can lead to wrong interpretations. In fact, the authors use a mortar in the experiments.

Also, the expression “structure interface” should be detailed.

It is not clear how the temperature is measured. At top/external face of the samples? This has influence on the results if temperature values change with the depth.

The results should be discussed by considering the previous publications related with the topic.

Specific comments

Line 38 - “tests for submerged geosynthetic-soil specimens under different chemical conditions”

This has nothing to do with the topic...

Line 132 - “The device can test specimens at precisely controlled temperatures of -20 °C – +30 °C through refrigeration and a temperature control system. When the specimen reaches the desired temperature, the vertical load is applied.”

Please, explain how and where the temperature values were taken.

Line 148 - “Warm frozen soil and ice-rich frozen soil have significantly different physical and mechanical properties and critically impact the structure’s safety.”

At this point we don’t have the results….

Line 155 - “The temperature oscillations for testing temperatures -1 °C and -14 °C are shown in Figure 3a”

Do you mean Fig 3b? Check this….

Line 179 - “are the forward peak shear stress and the absolute value of the reverse peak shear stress”

The forward/reverse stress was not explained in the methodological procedures.

Line 225 - Check the figure 5. Some parts of the graphs are missing. Please, add the meaning of the red/black lines.

Line 230 - “The anisotropy extent increases with increasing normal stress and decreasing temperature”

This is a rough approach. In fact, the pattern is different when a specific case (temperature or stress) is examined.

Line 238 - “The mechanism for variation in anisotropy extent with normal stress and temperature is the same as that for N = 1.”

When similar conditions are compared obvious differences stand out.

Line 251 - “The fitting curves for different temperatures are shown in Figure 8”

The data presented in Figs 5 and 8 is different…. Until 10th cycle 6 points are plotted in Fig 5, while 10 points are plotted in Fig 8.

Line 341 - “Figure 11. Surface images of the cemented sand specimens at different temperatures before and after testing.”

It is not clear which one of this samples is the before/after the  test.

Line 443 - “A similar phenomenon has been reported previously Error! Reference source not found.”

Check

Line 492 - “structure interfaceError! Reference source not found.”

Check

Author Response

The authors appreciate the comments from reviewers. The questions and comments are very encouraging, constructive, and helpful in improving the quality of this manuscript. Detailed response to each question is presented below. The additions/revisions in the manuscript are highlighted in red.

Round 2

Reviewer 2 Report

General comments

Despite the improvements made on the manuscript, the expression “warm and cold frozen cement-treated sand” is strange. In this research the shear behaviour of cement-treated sand is assessed at several freezing temperatures. The used expression doesn’t express correctly the idea.

The results should be discussed by considering the previous publications related with the topic. There are several publications about this topic.
